# Infants’ Gaze Patterns for Same-Race and Other-Race Faces, and the Other-Race Effect

**DOI:** 10.3390/brainsci10060331

**Published:** 2020-05-29

**Authors:** Anna Krasotkina, Antonia Götz, Barbara Höhle, Gudrun Schwarzer

**Affiliations:** 1Department of Developmental Psychology, Justus-Liebig-Universität Gießen, 35394 Giessen, Germany; Gudrun.Schwarzer@psychol.uni-giessen.de; 2Department Linguistics, Faculty of Human Sciences, University of Potsdam, 14469 Potsdam, Germany; antgoetz@uni-potsdam.de (A.G.); hoehle@uni-potsdam.de (B.H.)

**Keywords:** eye-tracking, infancy, habituation, other-race effect, face discrimination

## Abstract

The other-race effect (ORE) can be described as difficulties in discriminating between faces of ethnicities other than one’s own, and can already be observed at approximately 9 months of age. Recent studies also showed that infants visually explore same-and other-race faces differently. However, it is still unclear whether infants’ looking behavior for same- and other-race faces is related to their face discrimination abilities. To investigate this question we conducted a habituation–dishabituation experiment to examine Caucasian 9-month-old infants’ gaze behavior, and their discrimination of same- and other-race faces, using eye-tracking measurements. We found that infants looked longer at the eyes of same-race faces over the course of habituation, as compared to other-race faces. After habituation, infants demonstrated a clear other-race effect by successfully discriminating between same-race faces, but not other-race faces. Importantly, the infants’ ability to discriminate between same-race faces significantly correlated with their fixation time towards the eyes of same-race faces during habituation. Thus, our findings suggest that for infants old enough to begin exhibiting the ORE, gaze behavior during habituation is related to their ability to differentiate among same-race faces, compared to other-race faces.

## 1. Introduction

The other race effect (ORE) begins to manifest during the first year of life, and is characterized by less efficient processing of faces from ethnicities that one does not have sufficient exposure to, when compared to faces from their own ethnicity (for a review, see [1]). Although recent studies showed that infants look at same- and other-race faces differently (e.g., [2,3,4], and see [5] for an overview) the role of infants’ visual encoding of same- and other-race faces for face discrimination is still not fully understood. Therefore, the goal of our study was to investigate the extent to which infants’ looking behavior while habituating to same- and other-race faces, is connected to their ability to discriminate among these faces.

During the first year of life there is a sensitive period when the face recognition system is exposed to inputs consisting of many exemplars of faces [6]. These inputs are thought to calibrate the face recognition system such that the infant begins to examine faces in an optimal way, for distinguishing faces matching the template of the initial calibrating sample. Given that infants have been shown to predominantly see same-race faces [7], this input might tune the face perception system towards the optimal perception of same-race faces. This hypothesis was supported by evidence that Caucasian [8], African [9], and Asian [10] 3-month-old infants who grew up in mono-ethical environments showed a looking preference for same-race faces when shown together with other-race faces. In terms of discrimination abilities, at 3 months of age, Caucasian and Asian infants did not yet exhibit differences in discrimination ability for same- and other-race faces, and only displayed a looking preference for the faces they were habituated to, regardless of race. However, by the age of 9 months, infants were able to discriminate only between same-race faces [11,12,13].

Regarding the mechanism behind this divergence of discrimination abilities, there is increasing evidence from eye-tracking studies that infants display different fixation patterns when looking at same- versus other-race faces (e.g., [2,3,4,14]). For example, when looking at same-race faces, 6- to 10-month-old Caucasian infants showed a higher proportion of fixation time for the eyes and showed a lower proportion of fixation time for the mouth, with increasing age. However, no such developmental changes were observed in the fixations for other-race faces [3]. Xiao and colleagues [14] found the same patterns in 6- and 9-month-old Caucasian infants. The same study also showed that at 9 months of age, infants looked significantly longer at the eyes of same-race compared to other races faces, and less at the mouth of same-race, as compared to other race faces. Thus, we already have an indication that infants might use different looking behaviors when viewing same-race and other-race faces.

It is important to note, however, that the above studies used passive viewing tasks to record infants’ looking behavior but did not relate the looking behavior to discrimination abilities. Little is known about whether infants’ looking behavior is connected with their discrimination performance for same-race and other-race faces. To the best of our knowledge, there are only two studies that investigated this question. Gaither and colleagues [15] investigated 3-month-old Caucasian and Asian infants’ ability to discriminate among Caucasian and Asian faces, using a habituation–dishabituation procedure, combined with the recording of infants’ gaze behavior. Consistent with previous studies [11,12], infants of both ethnicities showed comparable discrimination abilities for both face categories. Nevertheless, the authors revealed significant positive correlations between same-race novelty looking preference and more frequent transitions between the eye and mouth regions of own-race habituation stimuli for Caucasian and Asian infants. A recent study [16] extended this research by testing 6- and 9-month-old Asian infants. After familiarization to same- and other-race face stimuli, infants in both age groups showed a novelty looking preference in the same-race but not in the other-race condition. Infants also showed different looking patterns for the same- and other-race faces. Both 6- and 9-month old infants looked significantly longer at the nose and mouth of same-race faces than of other-race faces, during familiarization. Additionally, the amount of time infants looked at the eyes and nose of faces during a recognition test was positively correlated with the recognition performance during the test trials. Taken together, these data suggest that 3-month-old Caucasian and Asian infants, as well as Asian infants between 6 and 9 months of age look differently at same- and other-race faces, during familiarization or habituation, and that these differences in their looking patterns are connected with their face discrimination performance.

The aim of the current study was to extend the above approach to determine whether this link between gaze behavior and discrimination ability also appears in Caucasian infants old enough to begin exhibiting the ORE. We therefore investigated whether 9-month-old Caucasian infants’ gaze behavior while habituating to same- and other-race faces is related to their discrimination of these faces. We tested only 9-month-old Caucasian infants because a previous study [16] revealed no age-related differences between 6- and 9-month old infants in the looking pattern for both same- and other-race faces, and the ORE was already observed at the age of 9 months in several studies [11,12,13]. As we wanted to replicate the previous findings regarding the ORE in infants, we used the same habituation–dishabituation procedure as [11,12,13]. This procedure allowed us to examine infants’ gaze behavior while they encode same- and other-race faces during the habituation phase, and enabled us to subsequently test the discrimination of the habituated faces from a novel face. We were then able to test whether the patterns of infants’ gaze behavior during habituation correlated with their discrimination abilities at test, in a similar manner to the study examining the gaze behavior and discrimination abilities of Asian infants [16].

Based on the finding of previous studies, we expected infants to show an ORE in terms of better discrimination of same-race faces, compared to other-race faces. We also anticipated Caucasian infants to use different looking patterns while habituating to same- and other-race faces. Based on the results of [4,14], we anticipated that infants might show differences in their fixation times towards the eyes, and perhaps the mouth of same-race faces, compared to other-race faces. Finally, we also expected to find a connection between infants’ looking patterns during habituation and their face discrimination performance based on the findings of [15,16].

## 2. Materials and Methods

### 2.1. Participants

We collected data from 68 healthy, full-term, Caucasian 9-month-old infants (*M* = 292.42 days; range: 275–304 days, 31 female and 37 male). To avoid any possible carry-over effects from the exposure to faces of varying races within the experiment, each infant participated in either the same-race or other-race condition, as in prior studies [11,12,13]. Thirty-four infants participated in the same-race condition, and the other 34 infants participated in the other-race condition. The sample size was chosen based on a statistical power analysis that used effects sizes taken from a study by [13], which found the ORE in 9-month-old infants with a comparable sample size. An additional 16 participants were excluded from the final sample because of fussiness (*n* = 5), failing the habituation criteria (*n* = 8), or technical problems during the experiment (*n* = 3). Infants were recruited through the municipal administration office. None of the infants included in our analyses had direct contact with individuals of Asian descent (no Asian relatives or acquaintances), according to a questionnaire administered to their parents.

### 2.2. Stimuli

Based on the results of a preliminary study (see below), we selected colored photographs of 6 Caucasian (German origin) and 6 Asian (Chinese origin) women for use as stimuli. Each woman was presented in three poses: frontal-facing, turned ¾ to the left, and turned ¾ to the right. In every photo, the woman looked directly at the camera while smiling, with the hair, neck, and shoulders visible. Smiling faces were chosen over faces showing neutral expressions to make the stimuli more appealing to the infants, as neutral faces have been shown to appear as emotionally negative to children [17]. Examples of stimuli pairs are shown in Figure 1.

To select our stimuli, we conducted a preliminary study with 15 German and 14 Chinese adults to ensure that the pairs of faces from the same ethnicity that were used for the test phase were of comparable similarity for the two ethnicities. We took 15 pictures with faces from each ethnicity (15 German women for the Caucasian category, and 15 Chinese women for the Asian category), combined them into pairs within each ethnicity, and showed them to the German and Asian adults. Participants rated each pair of photographs twice using a scale from 1 (very similar) to 9 (very different). German subjects rated the Caucasian German faces, and the Chinese subjects rated the Chinese Asian faces.

The three most similar pairs from each ethnicity category were then selected as the stimuli for our main study—6 Caucasian faces in three pairs (with average similarity ratings of 4.69, 4.69, and 4.81), and 6 Asian faces in three pairs (with average similarity ratings of 4.59, 4.73, and 4.73).

### 2.3. Procedure

Parents were informed regarding the study procedure and gave written consent for their child’s participation. All parents were blind to the study hypotheses. Infants sat on their parent’s lap approximately 60 cm from a computer monitor with a screen resolution of 1920 × 1080 pixels, and an integrated Tobii TX300 eye tracker with a data-sampling rate of 300 Hz. The nominal accuracy was given as 0.4°–0.5° at our lighting conditions (office-lighting), as per the TX300 manufacturer, and the nominal precision was given as 0.04°–0.06° using noise-reduction data filtering, although as shown by Hessels and Hooge [18], in practice such parameters could prove to be noticeably worse, especially when testing infants. See https://www.tobiipro.com/siteassets/tobii-pro/product-descriptions/tobii-pro-tx300-product-description.pdf for additional technical specifications of the Tobii TX300 system.

The parents were instructed to close their eyes and sit still during the experiment. In order to record infants’ gaze behavior, we used a 5-points calibration procedure (4 corners and the middle of the screen). The calibration procedure was repeated until calibration was achieved successfully for all 5 points, up to a maximum of 4 attempts (the data from participants who failed the calibration procedure were excluded from data analysis).

Participants were randomly assigned to one of two conditions (Caucasian or Asian faces). Each picture was displayed 12.5 cm (11.89°) wide and 16.5 cm (15.66°) high on the screen. We used an infant-controlled habituation-dishabituation procedure with a maximum of 18 habituation trials. During habituation, pictures of a single face were shown sequentially, randomly alternating between the three possible poses (photos were presented in sequences of three, with each sequence containing all three poses in a random order). An acoustic attention-getter was played with the appearance of each picture. Pictures were presented until the infant looked away for 2 s, or until the maximum trial length of 40 s was reached. We considered that an infant was habituated when the mean looking time of the three last habituation trials was less than 50% of the mean of the first three habituation trials. The test phase started immediately after this habituation criterion was reached, or after the maximum of 18 habituation trials were presented. Those infants who saw all 18 habituation trials without having reached the habituation criterion were considered to not have been habituated, and were excluded from further analysis. In the test phase, the previously habituated face was shown in a randomly alternating order with another face from the same race category as the habituated face (from the originally selected pair of faces), with both faces presented frontally (one at a time) for a maximum duration of 40 s, for a total of two test trials. E-Prime version 2.0 (Psychology Software Tools, Pittsburgh, PA, USA) was used to control the stimulus presentation.

### 2.4. Data Analysis

The eye-tracking data were analyzed with the Tobii Pro Studio using the included ClearView fixation filter using the following setting. Fixations were defined with a minimum looking period of 100 ms, within a radius of 30 pixels (0.765°) for the consecutive samples, based on previous analyses of eye tracking data from infants ([2,4]). The nominal precision (0.04°–0.06°) given by the manufacturer for the TX300 was below 0.1°, which according to examples demonstrated by Holmqvist and colleagues [19] should be sufficient for the minimal loss of fixations detected. We created five Areas of Interest (AOIs)—whole-head, left eye, right eye (during analysis, we combined the left and right eyes together into a single AOI), nose, and mouth. Figure 2 shows an example of AOI positioning. The size of each AOI was the same for each picture, and the exact position was individually adjusted for each face. For each AOI within (each trial), we calculated total dwell time as the sum of the durations in seconds of all fixations made by each particular infant in that AOI. Mean total dwell time for each participant was then calculated as the average of total dwell times across all trials of the habituation phase of the experiment. Similar to the analysis used by Liu and colleagues [16], proportional total dwell times were computed by dividing the mean total dwell times of the individual AOI’s (eyes, nose, mouth) by the mean total dwell time of the whole-head AOI.

Infant face discrimination was assessed via their novelty preference scores, which was defined as the total dwell time for the novel face (whole head AOI) as a proportion of the sum of the total dwell times for the novel and habituated faces at test. To look for connections between infants’ gaze behavior during habituation and their discrimination ability at test, we computed correlations between infants’ proportional total dwell times for the various AOIs during habituation, and their novelty preferences.

## 3. Results

### 3.1. Preliminary Analyses

To ensure that a similar level of eye-tracking data quality was obtained for both conditions, we conducted a one-way ANOVA on the percentages of the samples in which the eye-tracker successfully registered the position of the infants’ gaze on the monitor, with stimulus-race condition (same-race, other-race) as a between subjects factor. We did not find a significant effect of stimulus-race condition, *F*(1, 66) = 0.060, *p* = 0.807, indicating a similar level of data capture quality in the same-race and other-race conditions (*M* = 80.235% vs. *M* = 80.823%, respectively).

To ensure that the infants in the same-race and other-race conditions did not differ with respect to the number of trials seen during habituation and the total dwell time for the stimuli, we conducted two further one-way ANOVAs using stimulus-race condition (same-race, other-race) as a between subjects factor. The first ANOVA was conducted on the number of trials that the infants saw during habituation, and we found no significant effect of stimulus-race condition, *F*(1, 66) = 0.636, *p* = 0.428, indicating that the infants saw a similar number of habituation trials in the same-race and other-race conditions (*M* = 8.12 vs. *M* = 8.71, respectively). The second ANOVA was conducted on the total dwell times (in seconds) of the infants for the whole-head AOI during habituation. Again, we found no significant effect of stimulus-race condition, *F*(1, 66) = 0.077, *p* = 0.782, indicating that the infants showed a similar total dwell time for our face stimuli during habituation in the same-race and other-race conditions (*M* = 67.16 s vs. *M* = 71.39 s, respectively).

### 3.2. Face-Discrimination Performance

A one-way between subjects ANOVA was conducted to compare novelty preference scores (the whole-head AOI total dwell time for the novel face divided by the sum of the whole-head AOI total dwell times for the novel and habituated faces) for infants in the test trials for the same- and other-race face conditions. There was a significant difference between the novelty preference scores (Figure 3) in the two conditions, *F*(1, 66) = 10.610, *p* = 0.002, indicating a significantly higher average preference score in the same-race compared to the other-race condition. To investigate whether the novelty preference scores for same- and other-race faces were above chance, we used two post-hoc one-sample *t*-tests to compare the novelty preference scores against a chance-level novelty preference score of 0.50 for each condition. The results indicated that the infants’ preference scores in the same-race condition were significantly above chance, (*M* = 0.669 s, *SD* = 0.202; *t*(33) = 4.872, *p* < 0.001), indicating that they were able to discriminate between the same-race faces. By contrast, the preference scores in the other-race condition (*M* = 0.527 s, *SD* = 0.154) were not significantly different from chance, *t*(33) = 1.018 *p* = 0.316, indicating that the infants could not discriminate between the other-race faces. This pattern of results clearly demonstrated the ORE.

### 3.3. Gaze Behavior

#### 3.3.1. Proportional Total Dwell Time for Individual AOIs during the Habituation Phase

To analyze the gaze behavior of infants during habituation, we first calculated the proportional total dwell times for the eyes, nose, and mouth AOIs (in total for the entire habituation stage), as described earlier. Using those proportional total dwell times as the dependent measure, we conducted a repeated measures ANOVA with the AOI (eyes, nose, mouth) serving as a within-subject repeated measure factor, and the stimulus-race condition (same-race, other-race) as a between-subjects factor. Greenhouse–Geisser corrections were used due to a violation of sphericity. We detected a significant main effect of stimulus-race condition, *F*(1, 66) = 21.255, *p* < 0.001, as well as a significant main effect of AOI, *F*(1.45, 95.685) = 20.055, *p* < 0.001. Especially interesting for our study, the results also showed a significant interaction between AOI and the stimulus-race condition, *F*(1.45, 95.685) = 3.630, *p* = 0.044, which indicated that infants in the two conditions distributed their attention differently between the three AOIs.

To further analyze this interaction, we conducted three individual one-way ANOVAs on the proportional total dwell times for the eyes, nose, and mouth AOIs, with the stimulus-race serving as a between-subjects factor in a similar analysis procedure as that used by Liu and colleagues [16]. We found a significant difference between the stimulus-race conditions in the eyes AOI, *F*(1, 66) = 10.698, *p* = 0.002, with the infants showing a greater overall proportional total dwell times for the eyes in the same-race (*M* = 0.388, *SD* = 0.218) as compared to the other-race condition (*M* = 0.216, *SD* = 0.215). The remaining two ANOVAs on the proportional total dwell times for the mouth and nose AOIs did not show any significant differences between the stimulus-race conditions (Figure 4).

#### 3.3.2. Relations between Proportional Total Dwell Times during Habituation and Novelty Preferences

To examine the relationship between infants’ looking patterns during habituation and their discrimination performance at test, we calculated Pearson correlations between the infants’ novelty preference scores, and their proportional total dwell times for the eyes AOI during habituation. We found a significant difference between the same-race and other-race conditions, as per the ANOVA analysis described above (Figure 4). We found a significant positive correlation between infants’ novelty preferences in the same-race condition during testing and their proportional total dwell times for the eyes AOI during habituation, r = 0.380, *p* = 0.027. Thus, the more the infants in the same-race condition looked at the eyes AOI during the habituation stage, the better they could discriminate between novel and familiar faces during the test phase (Figure 5). By contrast, infants in the other-race condition showed no significant correlation between their novelty preferences during testing and their proportional total dwell times towards the eyes AOI during habituation, r = 0.008, *p* = 0.963. Using a Fischer r-to-z transformation, we compared the above correlation coefficients from the same-race and other-race conditions, which showed the difference to be marginally significant, *p* = 0.062.

## 4. Discussion

The aim of the present study was to investigate whether there are differences in the fixation patterns of 9-month-old Caucasian infants during habituation to same- and other-race faces, and whether these differences are connected to the ORE, i.e., better discrimination of same-race faces compared to other-race faces. Our results showed that the 9-month-old infants exhibited a clear ORE, as they showed a novelty preference in the Caucasian same-race condition, but not in the Asian other-race condition. Furthermore, our data showed that there are differences between the fixation patterns during habituation to Caucasian same-race faces versus Asian other-race faces. Infants showed greater proportional total dwell times (relative to the entire face) for the eye region of the same-race faces compared to other-race faces. Our findings also indicated that for the same-race faces, the proportion of time infants spend fixating the eyes AOI during habituation, significantly correlated with their discrimination abilities during testing. No such correlation was found for the other-race faces.

Our detection of the ORE in 9-month-olds confirmed results from numerous previous studies (e.g., [11,12,13,16]), showing that at the age of 9 months, infants begin to show a robust ORE if they have no direct contact with people from other ethnicities. Our findings also agreed with the results of previous eye-tracking studies on infants (e.g., [2,3,4,16]), showing that infants look differently at same- and other-race faces. In our study, Caucasian infants fixated significantly longer on the eyes of same-race faces than of the other-race faces, which agrees with the results of previous studies [3,4,14], where Caucasian infants also fixated more on the eyes of same-race faces than other-race faces. A preference for fixating on the eye area of more familiar face stimuli was previously observed in Caucasian infants as early as 3 months old. In one study [20], human infants fixated more on the eyes of human faces than on the eyes of monkey faces. A preference for looking at the eyes of same-race faces compared to other-race faces was also found in Caucasian adult participants [21,22,23]. However, unlike some studies such as [3,14], the infants in our study did not show significant differences in proportional total dwell times regarding the mouth region of same- and other-race faces. This difference between our results and those of the above studies could be explained by differences in stimulus materials. While [3,14] used videos of moving faces in their studies, we used pictures. The stillness of the faces in our study could have made them appear as less communicative, and thus attracted less attention towards the mouth, which is a key component in the perception of verbal communication.

Our results also agreed with previous infant studies that showed that there is a connection between infants’ gaze behavior while encoding faces and their performance in recognizing faces (e.g., [15]). In our study, we found that longer proportional total dwell time on the eyes AOI during habituation corresponds with better face discrimination in testing in the same-race condition, but not in other-race conditions. These results are, to our knowledge, the first to demonstrate that the gaze patterns of Caucasian infants old enough to exhibit the ORE are directly linked to their ability to discriminate between same-race and other-race faces. Thus, our findings suggest that the ORE at 9 months of age might at least partially stem from the lack of attention paid to the eye region of other-race faces by infants, and that infants might place less weight on the informative value of the eyes, with respect to the identity of other-race face during encoding, which could in turn lead to poorer other-race face discrimination.

One limitation of our study was the usage of still photographs rather than videos for face stimuli. An important step for future studies would be the production and usage of video stimuli, which would still conform to our requirements in terms of their visual parameters. Such video stimuli could add important new insights to the role of gaze patterns in the ORE, since such stimuli might offer additional useful visual cues for face discrimination. For example, a study by Xiao and colleagues [24] demonstrated that the recognition of moving faces seems to be more dependent on the quantity of infants’ fixation shifts than that of static faces.

An additional point worth noting is that although we demonstrated a correlation between the infants’ proportional total dwell times for the eyes during habituation and their discrimination ability, we cannot definitively prove causation. In order to prove that infants’ dwell time for the eyes has an effect on face discrimination, future studies would need to manipulate the information that infants can gain from the eye region and measure the resulting discrimination performance. One possible way to achieve this would be, for example, to show the eye region with varying levels of blurring during habituation.

Another aspect of our experiment which bears further consideration was the use of smiling faces as opposed to neutral faces for stimuli. The ORE was already demonstrated in 9-month-old infants using smiling faces [17], and so we chose to use smiling faces as well, in order to make the stimuli appear more pleasant for the infants. However, emotional expressions were recently shown to play a role in the ability of infants to recognize other-race faces. For instance, Quinn and colleagues [25] demonstrated that infants old enough to exhibit the ORE were able to differentiate other-race faces better when they displayed emotions such as happiness or anger, than when they showed a neutral expression. While the infants in the present study still displayed the ORE, even for the happy faces, this could be a result of the face pairs in our experiments being selected to be as similar as possible, and thus, being more challenging to discriminate. Nevertheless, it would be important to expand our investigation to encompass other facial expressions, including neutral expressions, in future studies in order to fully understand the roles of perceptual narrowing and emotional expressions in the ORE.

It is also important to note that we only tested Caucasian infants in our study. Although it was demonstrated that the ORE appears in all races [8,9,10,11,12], it was also shown that children from different races look at faces by using different gaze patterns [2,3,4,16]. It would thus be interesting to extend our findings to further cultures to see whether and how the differences in looking patterns we found during habituation, and their relation to discrimination abilities, might differ across cultures. Likewise, it would also be interesting to see if the looking behaviors we observed in Caucasian infants were common across stimuli of additional ethnicities, since we only used Asian faces for our other-race face stimuli.

## 5. Conclusions

Our findings suggest a connection between the ORE and the gaze patterns that 9-month-old Caucasian infants use when examining same- and other-race faces. Our results showed that during habituation infants fixated significantly longer on the eye region of the same-race faces compared with the other-race faces, as a proportion of overall fixation time towards the stimuli. We then found a significant correlation between the infants’ proportional total dwell time for the eyes of same-race faces during habituation and their discrimination of those faces during the test. By contrast, we found no such correlation for other-race faces. These findings offer novel evidence that directly links the gaze behavior of Caucasian infants during habituation to other-race effects during discrimination testing.

## Figures and Tables

**Figure 1 brainsci-10-00331-f001:**
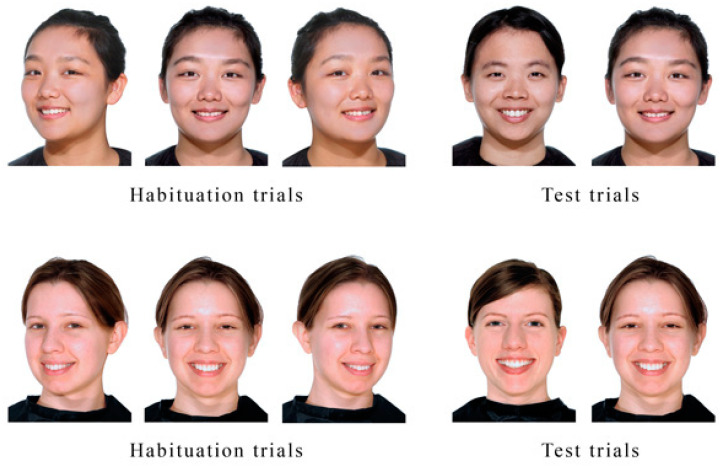
Examples of other-race (above) and same-race (below) faces used in the habituation and test trials. We are authorized to use and publish the photographs of the person in the figures.

**Figure 2 brainsci-10-00331-f002:**
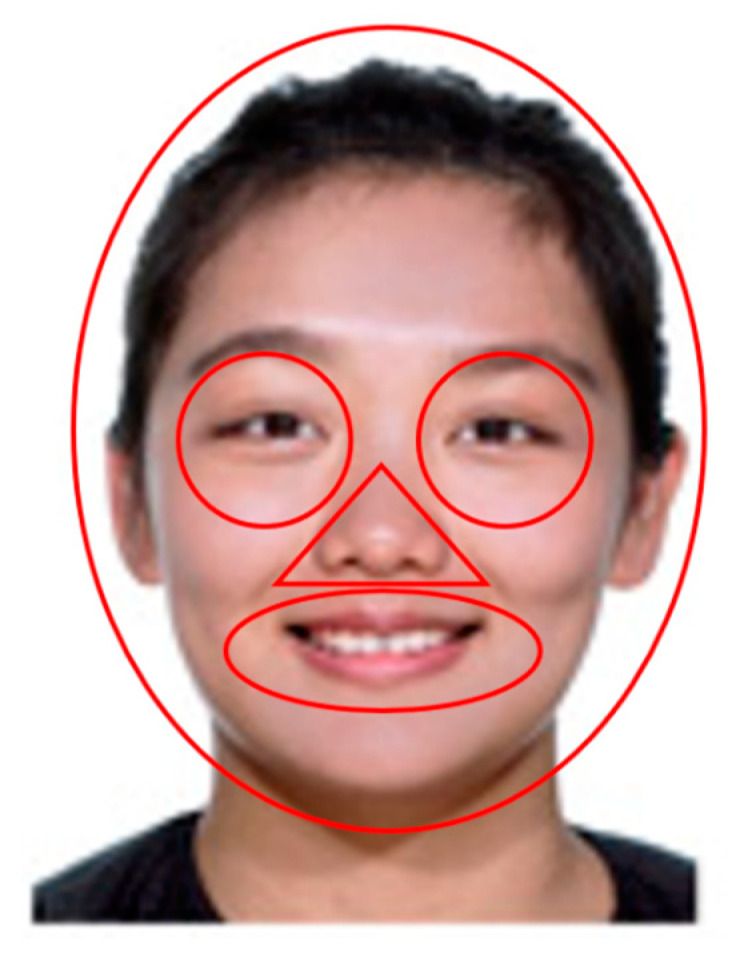
Illustration of areas of interest (AOIs) corresponding to the left eye (width: 104 pixels, 2.651°; height: 104 pixels, 2.651°), right eye (width: 104 pixels, 2.651°; height: 104 pixels, 2.651°), nose (width: 91 pixels, 2.320°; height: 55 pixels, 1.402°), mouth (width: 176 pixels, 4.485°; height: 87 pixels, 2.218°), and the whole-head (width: 399 pixels, 10.147°; height: 479 pixels, 12.167°). We are authorized to use and publish the photographs of the person in the figures.

**Figure 3 brainsci-10-00331-f003:**
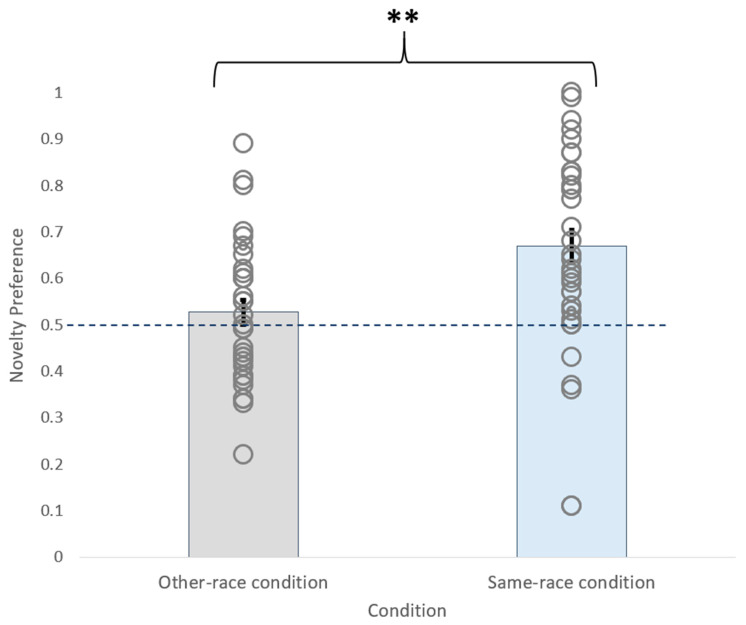
Novelty preference scores for the same-race and other-race conditions. Circles indicate the individual data points, and the dotted line indicates the expected chance-level novelty preference of 0.5. Error bars indicate standard error, and two stars indicate a significance level of *p* < 0.01.

**Figure 4 brainsci-10-00331-f004:**
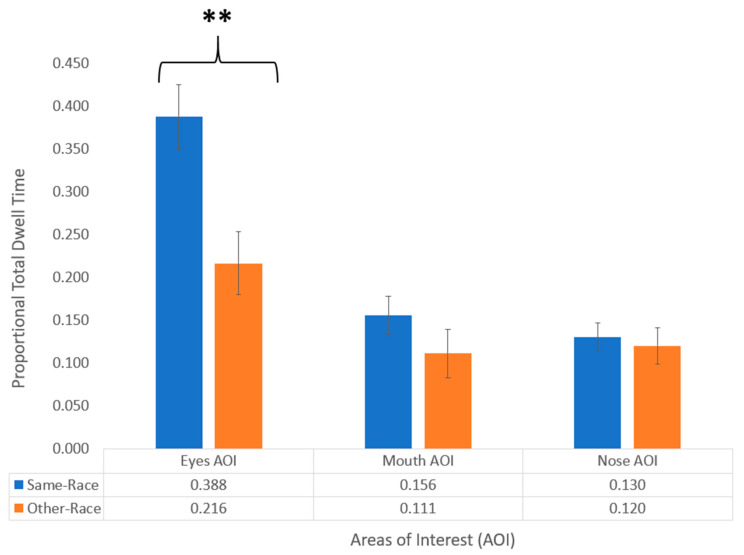
Proportional total dwell times for eyes, nose, and mouth AOIs during the habituation phase for the same-race and other-race conditions. Error bars indicate standard error, and the two stars indicate a significance level of *p* < 0.01.

**Figure 5 brainsci-10-00331-f005:**
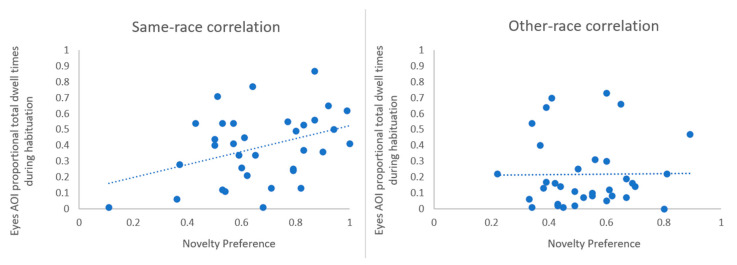
Correlation between infants’ novelty preference scores during testing and their proportional total dwell times for the eyes AOI during habituation in the same-race condition (**left**) and the other-race condition (**right**).

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
