# Peer review of "Infants’ Gaze Patterns for Same-Race and Other-Race Faces, and the Other-Race Effect"

_brainsci, 2020, doi:10.3390/brainsci10060331_

Round 1
Reviewer 1 Report
This paper is fine, nothing groundbreaking, but also not bad. I think it should be published, at least somewhere, and I appreciate that doing infant studies is likely hard and time-consuming. The paper is short, simple, and straightforward, which is kind of refreshing. This also means that there is a relative lack of talk about theory, but it is up to the editor to decide whether that really matters – perhaps not every single paper on the ORE needs to discuss theoretical context in excruciating detail. Below are my more specific comments:
Figure 1. I worry about the role of extra-facial cues, e.g. the two Caucasian faces have clearly different hairstyles while the two Asian faces do not have drastically different hairstyles.
Line 122: It was a good idea to do the preliminary study, but why did you not ask people of both ethnicities rate all images? I worry about anchoring effects, where the similarity ratings are not fully comparable as they ratings might be done only relative to the faces within the same face set, i.e. within the same ethnicity.
Line 132: “The three most similar pairs from each ethnicity category (6 Caucasian faces, and 6 Asian faces) were then selected as the stimuli for our main study.” What were their similarity ratings?
Figure 3: This figure tells the reader nothing that he/she cannot get from the main text as it only shows means. The figure legend also talks about two stars, but no stars are shown. The fill patterns are also distracting, and there is no need to have a header for the plot – that belongs in the figure legend. Please remake this figure and show individual data points, or at least the spread of data. I suggest bean plots or similar. Indicate chance level by a dotted line or similar. I also suggest getting rid of grid lines.
Line 202: “...three individual one-way ANOVAs on these fixation proportions for the eyes, nose, and mouth AOIs, with stimulus-race serving as a between-subjects factor.” Why not do a single ANOVA with two factors, one of which is AOI, and then look for an interaction between AOI and stimulus-race?
Figure 4: Similar comments to those for figure 2.
Line 215: “…we calculated Pearson correlations between the infants’ novelty preference scores and their fixation proportions for the eyes AOI during habituation where we found a significant difference between the same-race and other-race conditions.” As far as I could understand, you did not compare these directly so you cannot make this claim until you do. What you did find was a significant relationship for same-race and a non-significant relationship for other-race. That in and of itself does not prove that the two _themselves_ are significantly different. You need to do that direct test.
Figure 5: The figures are stretched, and they are also duplicated. Again, I suggest, getting rid of grid lines.
Line 273: Funding should be finding.
Author Response
Figure 1. I worry about the role of extra-facial cues, e.g. the two Caucasian faces have clearly different hairstyles while the two Asian faces do not have drastically different hairstyles.
That is a fair point. However, since we were interested primarily in how infants’ look at the invariant internal face features typical for this type of research (i.e. the eyes, nose, and mouth), and selected our stimulus pairs such that they would have very similar ratings of within-pair similarity (as rated by same-race adults to avoid other-race effects interfering at this point), we decided to leave the hair in place as it would make the stimuli appear more natural to the infants.
Line 122: It was a good idea to do the preliminary study, but why did you not ask people of both ethnicities rate all images? I worry about anchoring effects, where the similarity ratings are not fully comparable as they ratings might be done only relative to the faces within the same face set, i.e. within the same ethnicity.
As mentioned above, the reason that we used similarity ratings provided by raters of the same race as the stimuli, was in order to minimize other-race effects in the similarity ratings, which could skew the perceived similarity of the faces for someone of a different ethnicity. We even made sure that the raters were from the same country of origin as the models in the photographs (i.e. German people rated the German faces, and Chinese people rated the Chinese faces).
Line 132: “The three most similar pairs from each ethnicity category (6 Caucasian faces, and 6 Asian faces) were then selected as the stimuli for our main study.” What were their similarity ratings?
Thank you very much for the suggestion. We have now added the individual similarity ratings for each pair instead of the average values of all the pairs.
Figure 3: This figure tells the reader nothing that he/she cannot get from the main text as it only shows means. The figure legend also talks about two stars, but no stars are shown. The fill patterns are also distracting, and there is no need to have a header for the plot – that belongs in the figure legend. Please remake this figure and show individual data points, or at least the spread of data. I suggest bean plots or similar. Indicate chance level by a dotted line or similar. I also suggest getting rid of grid lines.
We deeply apologize for the confusion, but for some reason all the mentioned elements which were originally present in the figures do not appear in the version that you received. We will make sure that all these elements of the figures appear properly in the revision. We also removed the figure headers, got rid of the fill patterns, and added the individual data points and a chance-level indicator, as you suggested
Line 202: “...three individual one-way ANOVAs on these fixation proportions for the eyes, nose, and mouth AOIs, with stimulus-race serving as a between-subjects factor.” Why not do a single ANOVA with two factors, one of which is AOI, and then look for an interaction between AOI and stimulus-race?
Originally we were primarily interested in the difference between the conditions for the individual AOIs, so we would have needed to conduct the individual analyses per AOI in any case, so we did those right away.
We have now conducted and added to our manuscript a single large ANOVA which included all AOIs and conditions at the request of one of the other reviewers. This ANOVA shows a significant interaction between condition and AOI, and so the three individual ANOVAs now serve as post-hoc tests for this interaction. Thank you for the suggestion!
Figure 4: Similar comments to those for figure 2.
Once again, we apologize for the problems issues with the figure and will make sure that all these elements of the figures appear properly in the revision.
Line 215: “…we calculated Pearson correlations between the infants’ novelty preference scores and their fixation proportions for the eyes AOI during habituation where we found a significant difference between the same-race and other-race conditions.” As far as I could understand, you did not compare these directly so you cannot make this claim until you do. What you did find was a significant relationship for same-race and a non-significant relationship for other-race. That in and of itself does not prove that the two _themselves_ are significantly different. You need to do that direct test.
Thank you for the suggestion! We compared the two correlation coefficients using the Fischer r-to-z transformation, and added the result to the section where we report the correlations.
Figure 5: The figures are stretched, and they are also duplicated. Again, I suggest, getting rid of grid lines.
Again, we apologize for the problems issues with the figure and will make sure that all these elements of the figures appear properly in the revision. We also got rid of the grid lines as you suggested.
Line 273: Funding should be finding.
Thank you for catching that!
Reviewer 2 Report
In this paper, the authors investigate gaze behaviour of 9-month old Caucasian infants in relation to their discrimination performance of own- and other-race faces. Overall, the paper is easy to read and succinctly written, and provides relevant empirical contribution to the literature on own- and other-race face processing. Yet, I have a number of concerns and questions (particularly regarding the methods and results) that the authors should address prior to publication. Finally, I have some minor comments that the authors might wish to take into account in a revision.
MAJOR:
1. Throughout the manuscript, the authors use terms such as “Looking proportion”, “Looking time proportion”, “Proportional fixation time”, “Fixation proportions”, etc. to refer to the same thing. I think this using these terms interchangeably is confusing. Moreover, the terms themselves are incoherent with the eye-tracking literature. I suggest that the authors stick to the terminology suggested by Holmqvist, K., Nyström, M., Andersson, R., Dewhurst, R., Jarodzka, H., & Van de Weijer, J. (2011). Eye tracking: A comprehensive guide to methods and measures. OUP Oxford.
In the case of the present paper, that would mean using the terms “total dwell time” to refer to the total duration that a particular AOI was looked at, and “relative total dwell time” to express the total dwell time to any AOI as a proportion of some combination of total dwell times. Importantly, “fixation duration” is commonly used to denote the average duration of fixations, which the authors do not report in their paper. Please revise the paper accordingly.
2. On l161 the authors give an operationalization of fixations. I suggest the authors also clarify their definition per the advice of:
Hessels, R. S., Niehorster, D. C., Nyström, M., Andersson, R., & Hooge, I. T. (2018). Is the eye-movement field confused about fixations and saccades? A survey among 124 researchers. Royal Society open science, 5(8), 180502.
3. The authors operationalize fixations using a dispersion threshold of 30 pixels. Please provide this value in degrees as well, and relate it to the quality of the eye-tracking data obtained, in particular the precision, see:
Holmqvist, K., Nyström, M., & Mulvey, F. (2012, March). Eye tracker data quality: what it is and how to measure it. In Proceedings of the symposium on eye tracking research and applications (pp. 45-52).
The reason that this is important, is that it is well established that eye-tracking data quality is often low in infant research, which may affect the measures one derives from the eye-tracking data (e.g. total dwell times, fixation durations, etc.). It is crucial that the authors verify that the eye-tracking data quality did not differ between the two groups in their experiment, such that any differences between these groups may not be attributed to the eye-tracking data, instead of the eye-movement behavior. See e.g. the paper below for a discussion of the causes and potential consequences:Hessels, Roy S., Diederick C. Niehorster, Chantal Kemner, and Ignace TC Hooge. "Noise-robust fixation detection in eye movement data: Identification by two-means clustering (I2MC)." Behavior research methods 49, no. 5 (2017): 1802-1823.
In the context of the authors’ paper, it may be that if eye-tracking data quality was particularly low for some infants, high noise and short periods of data loss may lead to fixations not being classified, which in turn results in lower total dwell times (and potentially shifted relative total dwell times to the eyes, nose, and mouth AOIs).
I also ask that the authors report the AOI sizes in degrees, and relate it to the accuracy of their measurements (assuming they have empirical values for this).
4. Figures 3 and 4 are not informative in terms of the distributions of the eye-tracking measures across infants (not to mention that the red line and significance stars mentioned in the caption of Figure 3 are not visible). I suggest the authors change their visualisations to adequately capture the distributions of the novelty preference and relative total dwell times across infants.The following paper has many suggestions on how to do this adequately:Rousselet, G. A., Pernet, C. R., & Wilcox, R. R. (2017). Beyond differences in means: robust graphical methods to compare two groups in neuroscience. European Journal of Neuroscience, 46(2), 1738-1748.
5. Regarding the statistical analyses of the relative total dwell times to the eyes, nose, and mouth, I wonder why 3 ANOVAs were conducted and not 1? Moreover, were the assumptions of normality met?
6. As the authors report, the “Looking time proportions” for eyes, mouth, and nose are given as a proportion of the total dwell time to the face. For other-race faces, this proportion is consistently lower than for same-race faces (see Figure 4). This means that these infants were looking somewhere else in the face. My question is: where did the infants look then? This is important to know: if, for example, the infants tended to look just outside the eyes AOI, it may be that the AOIs did not capture the ‘eye’-region for other-race faces adequately.
7. In Figure 5 the same graphs are presented twice (top and bottom row). Moreover, the Figures are stretched in vertical direction.
MINOR:
l49: “increased” sounds like the infants actively decided to do this. Could this be rephrased in a less active manner?
l66: “increased scanning frequency between eyes and mouth”: While I understand what the authors mean by this sentence, “scanning frequency” does not seem apt. I suggest the word “transition”.
l75-76: The authors state that the infants “use different looking patterns”. Again, this seems like an active choice made by the infant (i.e. a strategy). Could this be phrased less actively, i.e. “the infants look differently at…”?
l141: “The calibration procedure was repeated until calibration was achieved successfully for all 5 points”. What was the criterion for “successful”?
l197: The section is titled “Eye movement results”. However, the analyses are not about eye movements, but rather about fixation locations (which are often defined as the absence of movement of a gaze location in space).
l263-266: Have the authors considered the alternative, namely that increased looking at the eyes is a result of better discrimination for the features that make up own-race faces? One could argue either way.
Typos/grammar:
l101: mean age is given in German number format (comma for decimals), should be UK/US number format I guess.
l102: dash after 304 should be removed.
l192: two points instead of one.
l252: This sentence seems grammatically incorrect.
Author Response
MAJOR:
- Throughout the manuscript, the authors use terms such as “Looking proportion”, “Looking time proportion”, “Proportional fixation time”, “Fixation proportions”, etc. to refer to the same thing. I think this using these terms interchangeably is confusing. Moreover, the terms themselves are incoherent with the eye-tracking literature. I suggest that the authors stick to the terminology suggested by Holmqvist, K., Nyström, M., Andersson, R., Dewhurst, R., Jarodzka, H., & Van de Weijer, J. (2011). Eye tracking: A comprehensive guide to methods and measures. OUP Oxford.
In the case of the present paper, that would mean using the terms “total dwell time” to refer to the total duration that a particular AOI was looked at, and “relative total dwell time” to express the total dwell time to any AOI as a proportion of some combination of total dwell times. Importantly, “fixation duration” is commonly used to denote the average duration of fixations, which the authors do not report in their paper. Please revise the paper accordingly.
Thank you very much for the suggestion! We decided to standardize on the term dwell time as you suggested, and used it to replace all the other terms in the relevant places in our manuscript.
- On l161 the authors give an operationalization of fixations. I suggest the authors also clarify their definition per the advice of:
Hessels, R. S., Niehorster, D. C., Nyström, M., Andersson, R., & Hooge, I. T. (2018). Is the eye-movement field confused about fixations and saccades? A survey among 124 researchers. Royal Society open science, 5(8), 180502.
Thank you for the suggestion, we added additional details regarding the fixation filter used, as per the suggestions in this paper.
- The authors operationalize fixations using a dispersion threshold of 30 pixels. Please provide this value in degrees as well, and relate it to the quality of the eye-tracking data obtained, in particular the precision, see:
Holmqvist, K., Nyström, M., & Mulvey, F. (2012, March). Eye tracker data quality: what it is and how to measure it. In Proceedings of the symposium on eye tracking research and applications (pp. 45-52).
Thank you for the suggestion, we have now added all measurements in degrees in addition to pixels, and added additional details concerning the accuracy and precision of our eyetracking system and how they relate to the settings used for detecting fixations.
The reason that this is important, is that it is well established that eye-tracking data quality is often low in infant research, which may affect the measures one derives from the eye-tracking data (e.g. total dwell times, fixation durations, etc.). It is crucial that the authors verify that the eye-tracking data quality did not differ between the two groups in their experiment, such that any differences between these groups may not be attributed to the eye-tracking data, instead of the eye-movement behavior. See e.g. the paper below for a discussion of the causes and potential consequences:Hessels, Roy S., Diederick C. Niehorster, Chantal Kemner, and Ignace TC Hooge. "Noise-robust fixation detection in eye movement data: Identification by two-means clustering (I2MC)." Behavior research methods 49, no. 5 (2017): 1802-1823.
In the context of the authors’ paper, it may be that if eye-tracking data quality was particularly low for some infants, high noise and short periods of data loss may lead to fixations not being classified, which in turn results in lower total dwell times (and potentially shifted relative total dwell times to the eyes, nose, and mouth AOIs).
We have now conducted an analysis comparing the eyetracking quality of the two groups and added it to our results section as per your suggestion.
I also ask that the authors report the AOI sizes in degrees, and relate it to the accuracy of their measurements (assuming they have empirical values for this).
Thank you for the suggestion! We have now added the dimensions of the AOI in pixels as well as degrees.
- Figures 3 and 4 are not informative in terms of the distributions of the eye-tracking measures across infants (not to mention that the red line and significance stars mentioned in the caption of Figure 3 are not visible). I suggest the authors change their visualisations to adequately capture the distributions of the novelty preference and relative total dwell times across infants.The following paper has many suggestions on how to do this adequately:Rousselet, G. A., Pernet, C. R., & Wilcox, R. R. (2017). Beyond differences in means: robust graphical methods to compare two groups in neuroscience. European Journal of Neuroscience, 46(2), 1738-1748.
Unfortunately there were technical errors in transferring our figures to the manuscript copy which you received, and many elements of the figures were missing. We apologize for these issues and will make sure that all these elements of the figures appear properly in the revision. We also made a number of changes to the figures in response to feedback from this review which we hope will be to your liking.
- Regarding the statistical analyses of the relative total dwell times to the eyes, nose, and mouth, I wonder why 3 ANOVAs were conducted and not 1? Moreover, were the assumptions of normality met?
We conducted individual ANOVAs for the 3 AOIs because we were primarily interested in differences between the conditions regarding the amount of visual attention that infants devote to those features during habituation. Therefore, we would have had to do individual analyses for each AOI eventually anyways, and so we decided to conduct those analyses right away as they addressed our main questions regarding infants’ gaze distribution during habituation. Assumptions of normality were met for all ANOVAs.
Additionally, we have now conducted and added to our manuscript a single large ANOVA which included all AOIs and conditions at the request of one of the other reviewers. This ANOVA shows a significant interaction between condition and AOI, and so the three individual ANOVAs now serve as post-hoc tests for this interaction.
- As the authors report, the “Looking time proportions” for eyes, mouth, and nose are given as a proportion of the total dwell time to the face. For other-race faces, this proportion is consistently lower than for same-race faces (see Figure 4). This means that these infants were looking somewhere else in the face. My question is: where did the infants look then? This is important to know: if, for example, the infants tended to look just outside the eyes AOI, it may be that the AOIs did not capture the ‘eye’-region for other-race faces adequately.
As illustrated in figure 2, our AOIs extend a little outward from the features they cover in order to capture fixations that were made close to a feature but were not exactly on point, and so we feel that fixations that fall outside of those broad AOIs cannot be fairly attributed to those features. Regarding those fixations that were made towards the faces, but which do not fall into the eyes, nose, or mouth AOIs, they are difficult to classify as they cannot be mapped to these specific features, and so we decided not to analyze them further as this would be too speculative.
- In Figure 5 the same graphs are presented twice (top and bottom row). Moreover, the Figures are stretched in vertical direction.
As mentioned earlier, there were unfortunately technical errors in the manuscript you saw regarding how the figures are displayed, and we will make sure that everything is displayed properly in the revision.
MINOR:
l49: “increased” sounds like the infants actively decided to do this. Could this be rephrased in a less active manner?
Thank you for the suggestion. We have now replaced the words “increased their proportion of fixation time” with “showed a higher proportion of fixation time”, and made a similar replacement later in the sentence.
l66: “increased scanning frequency between eyes and mouth”: While I understand what the authors mean by this sentence, “scanning frequency” does not seem apt. I suggest the word “transition”.
We have replaced the words ”increased scanning frequency” with “more frequent transitions” as you suggested.
l75-76: The authors state that the infants “use different looking patterns”. Again, this seems like an active choice made by the infant (i.e. a strategy). Could this be phrased less actively, i.e. “the infants look differently at…”?
We changed this phrasing exactly as you suggested.
l141: “The calibration procedure was repeated until calibration was achieved successfully for all 5 points”. What was the criterion for “successful”?
The Tobii Studio software which controlled our eyetracker, indicated whether or not it had collected sufficient data for each of the 5 calibration points, and displayed the actual positions of the recorded fixations relative to the animated attention getter which appeared at each of the 5 points. If any point had insufficient data, or if the recorded fixations for that point landed further away from the circular border of the attention getter than the diameter of that attention getter we would recalibrate the infant for that point.
l197: The section is titled “Eye movement results”. However, the analyses are not about eye movements, but rather about fixation locations (which are often defined as the absence of movement of a gaze location in space).
Thank you for pointing that out, we rephrased this subtitle to “Gaze position results”.
l263-266: Have the authors considered the alternative, namely that increased looking at the eyes is a result of better discrimination for the features that make up own-race faces? One could argue either way.
Thank you for the suggestion, since our correlation result indeed cannot allow us to definitely claim this as a causation, we have now added a brief discussion of this point in our discussion.
Typos/grammar:
l101: mean age is given in German number format (comma for decimals), should be UK/US number format I guess.
l102: dash after 304 should be removed.
l192: two points instead of one.
l252: This sentence seems grammatically incorrect.
All these small errors have now been correct in the text, thank you for pointing them out!
Reviewer 3 Report
The authors investigated whether Caucasian 9-month-olds’ looking behavior toward same-race and other-race faces during habituation differed and whether these differences correlated with novelty preferences during the test trials. They found that the infants allocated a greater percentage of attention toward the eyes of the same-race than other-race faces, but percentage of attention toward the nose and mouth regions did not significantly differ. Replicating previous findings, they found the 9-month-olds discriminated between same-race novel and familiar faces but not between other-race novel and familiar faces. Moreover, same-race discrimination ability correlated with percentage of time directed toward the eyes of same-race faces during habituation.
In general, the paper is well written. The authors might consider adding Xiao et al. (2015) to the introduction because that paper examined how scanning behavior during familiarization was related to face discrimination. The method is appropriate and designed to be comparable to previous research, enabling comparison of results to similar studies utilizing different samples and stimuli. With that stated, the authors should explain why they utilized smiling faces rather than neutral expression faces given most of the research assessing the ORE in infants has used neutral facial stimuli.
My suggestions are mainly to provide more data for the reader. For example, there are no data regarding the average and range of trials it took for infants to habituate and whether trials to habituate differed based on whether infants viewed same-race or other-race faces. Moreover, what was the average overall looking time toward the faces during habituation and did that differ based on face race? Also, habituation entailed seeing the familiarized face in three different poses and it would be useful to know whether attention to the various AOIs (eyes, nose, and mouth) differed based on pose or was similar regardless of pose. Therefore, an analysis examining how face race and pose contribute to attention toward the various AOIs and looking time during a trial would be useful and provide more in-depth information regarding encoding during habituation.
Because the test faces were always presented frontally, it would also be useful to know whether attention toward the eye AOIs for frontal poses during habituation more strongly correlated with novelty preferences at test than correlations between novelty preferences and attention toward the eye AOIs for ¾ left and ¾ right poses. It would be useful to assess these correlations for the nose and mouth AOIs as well.
The discussion could be strengthened by indicating what the results from this study uniquely contribute to the literature. The majority of the discussion focuses on how these results are similar to previous research and although I think replication is incredibly important, it is also critical to tell the reader what this work adds to our understanding of face recognition in general and the ORE in particular.
Another point that needs to be discussed is that the authors used smiling faces for their stimuli whereas the majority of research has utilized neutral expression faces. In a recent article, Quinn et al. (2020) found that 6-month-old Caucasian infants recognized Asian female faces when they were familiarized to and tested with those faces posing happy expressions, but not neutral expressions. Similarly, 9-month-old Caucasian infants recognized Black female faces when they were familiarized to and tested with those faces posing happy expressions, but not neutral expressions. The results from the current study do not replicate those findings (9-month-olds had difficulty recognizing other-race faces with smiling expressions) and those differences need to be considered and discussed.
Minor Comments
Line 165 on p. 4 and Line 201 on p. 6 – Change “Similarly” to “Similar”
Figure 5 on p. 7 is shown twice.
p/ 8 - There are xxtra spaces between “robust” and “ORE” on line 242 and between “with” and “the” on line 246
p. 8/Line 248 – remove “being” from the end of the line
p. 8/line 249 – remove “at” from the middle of the sentence
Author Response
In general, the paper is well written. The authors might consider adding Xiao et al. (2015) to the introduction because that paper examined how scanning behavior during familiarization was related to face discrimination. The method is appropriate and designed to be comparable to previous research, enabling comparison of results to similar studies utilizing different samples and stimuli. With that stated, the authors should explain why they utilized smiling faces rather than neutral expression faces given most of the research assessing the ORE in infants has used neutral facial stimuli.
Thank you for the suggestion! We added the paper you mentioned to the discussion where we talk about the importance of using video stimuli in future studies on our topic, like Xiao and colleagues did in theirs. We also clarified in the methods section that smiling faces were used to make the stimuli more pleasant for the infants because neutral faces may actually be perceived as negative by children.
My suggestions are mainly to provide more data for the reader. For example, there are no data regarding the average and range of trials it took for infants to habituate and whether trials to habituate differed based on whether infants viewed same-race or other-race faces. Moreover, what was the average overall looking time toward the faces during habituation and did that differ based on face race? Also, habituation entailed seeing the familiarized face in three different poses and it would be useful to know whether attention to the various AOIs (eyes, nose, and mouth) differed based on pose or was similar regardless of pose. Therefore, an analysis examining how face race and pose contribute to attention toward the various AOIs and looking time during a trial would be useful and provide more in-depth information regarding encoding during habituation.
Thank you for the suggestion. We have now added data regarding the number of habituation trials that infants saw in each condition, as well as the total fixation durations over the course of habituation for our faces in the two conditions. We also analyzed the differences in those values with respect to the same-race and other conditions, and added the results of this analysis to a new verification subsection of the Results section of the manuscript.
Regarding the poses, that is a very interesting point! The question of how viewpoint plays into the habituation and discrimination of faces during infancy is a fascinating one. Unfortunately, since we were focused primarily on the influence of the stimulus race on the infants' gaze, we used the three poses during habituation as a way of making sure that infants were habituated to a particular face, rather than a particular picture. In other words, it was something that we controlled (by ensuring that the 3 poses were evenly distributed throughout the habituation phase for both conditions) in order to make sure that it did not affect our ability to interpret our data with regard to our primary question. And since viewpoint was included in our method as something we controlled rather than manipulating it as an experimental variable, we did not track it in our eyetracking data output files. It might possible to piece that data back together from presentation log files, but it would take a long time to do so and to combine that information with the eyetracking data. However, since it does sound like an interesting question, we may attempt to go through this process after the publication of the current study, and if we are successful in recovering that data we will certainly try to publish the results of such an analysis.
Because the test faces were always presented frontally, it would also be useful to know whether attention toward the eye AOIs for frontal poses during habituation more strongly correlated with novelty preferences at test than correlations between novelty preferences and attention toward the eye AOIs for ¾ left and ¾ right poses. It would be useful to assess these correlations for the nose and mouth AOIs as well.
As mentioned above, the exact pose used in each individual trial was controlled to ensure even distribution, but was not tracked in the output data. We will do our best to reconstruct this data for analysis in a future study and can try to keep you up to date if you are interested in the outcome!
The discussion could be strengthened by indicating what the results from this study uniquely contribute to the literature. The majority of the discussion focuses on how these results are similar to previous research and although I think replication is incredibly important, it is also critical to tell the reader what this work adds to our understanding of face recognition in general and the ORE in particular.
Thank you for the suggestion, we added to our discussion to more clearly highlight the novel contributions of our study.
Another point that needs to be discussed is that the authors used smiling faces for their stimuli whereas the majority of research has utilized neutral expression faces. In a recent article, Quinn et al. (2020) found that 6-month-old Caucasian infants recognized Asian female faces when they were familiarized to and tested with those faces posing happy expressions, but not neutral expressions. Similarly, 9-month-old Caucasian infants recognized Black female faces when they were familiarized to and tested with those faces posing happy expressions, but not neutral expressions. The results from the current study do not replicate those findings (9-month-olds had difficulty recognizing other-race faces with smiling expressions) and those differences need to be considered and discussed.
That is a good point, and we have now added a discussion of this issue to our discussion section.
Minor Comments
Line 165 on p. 4 and Line 201 on p. 6 – Change “Similarly” to “Similar”
Figure 5 on p. 7 is shown twice.
p/ 8 - There are xxtra spaces between “robust” and “ORE” on line 242 and between “with” and “the” on line 246
- 8/Line 248 – remove “being” from the end of the line
- 8/line 249 – remove “at” from the middle of the sentence
Thank you for catching all these typos and errors, we fixed them all in the revision. The problem with the figure is unfortunately an error in the way that the figures are displayed. We will double check with the editor to make sure that all the figures appear correctly in the revision.
Round 2
Reviewer 2 Report
I thank the authors for their careful revisions in response to the comments I raised on the previous version of this manuscript. The authors have addressed most of my concerns satisfactorily. I have two comments remaining that pertain to my previous comments:
1. The authors have adapted the term 'dwell time' throughout the manuscript. However, it seems 'total dwell time' (or 'relative total dwell time' for the relative measures) are more appropriate. 'Dwell time' is normally used to denote the time from entry to exit of an AOI (i.e. one look to an AOI), whereas the total dwell time represents the sum of these individual dwells.
2. The description of eye-tracking data quality in the 'Data analysis' section does not seem quite right. A couple of concerns:
- Accuracy is generally not problematic for fixation classification, as it represent the systematic shift between the actual and reported gaze locations. Rather, accuracy is relevant for assigning fixations to AOIs. Thus, the accuracy should be compared to AOI sizes not to the fixation classification threshold.
- Both the accuracy and precision values are reported based on the manufacturer specifications. However, it has clearly been established that in infant research, these values are hardly ever obtained (see Hessels & Hooge (2019). Eye tracking in developmental cognitive neuroscience–The good, the bad and the ugly. Developmental cognitive neuroscience, 40, 100710). These values can be up to an order of magnitude larger than what is reported by the manufacturers (and here reported by the authors). Ideally, empirical values for accuracy and precision should be reported and compared between groups. At minimum, the authors should note that the values they report are manufacturer specifications, but that these values are generally higher for infants and thus likely also in the present study.
Author Response
- The authors have adapted the term 'dwell time' throughout the manuscript. However, it seems 'total dwell time' (or 'relative total dwell time' for the relative measures) are more appropriate. 'Dwell time' is normally used to denote the time from entry to exit of an AOI (i.e. one look to an AOI), whereas the total dwell time represents the sum of these individual dwells.
Thank you for the suggestion, we implemented this change throughout our manuscript.
- The description of eye-tracking data quality in the 'Data analysis' section does not seem quite right. A couple of concerns:
- Accuracy is generally not problematic for fixation classification, as it represent the systematic shift between the actual and reported gaze locations. Rather, accuracy is relevant for assigning fixations to AOIs. Thus, the accuracy should be compared to AOI sizes not to the fixation classification threshold.
- Both the accuracy and precision values are reported based on the manufacturer specifications. However, it has clearly been established that in infant research, these values are hardly ever obtained (see Hessels & Hooge (2019). Eye tracking in developmental cognitive neuroscience–The good, the bad and the ugly. Developmental cognitive neuroscience, 40, 100710). These values can be up to an order of magnitude larger than what is reported by the manufacturers (and here reported by the authors). Ideally, empirical values for accuracy and precision should be reported and compared between groups. At minimum, the authors should note that the values they report are manufacturer specifications, but that these values are generally higher for infants and thus likely also in the present study.
Thank you for pointing out these two issues. We removed the sentence which speaks about accuracy as it relates to the fixation radius since it is not appropriate, and added a caveat to the manufacturer specifications that in practice these values can be noticeably worse, especially when testing infants.
Reviewer 3 Report
The authors have addressed all my previous comments satisfactorily. I realize that the request to examine correlations between novelty preferences and attention toward AOIs as a function of familiarization pose is not possible due to the way the study was developed and/or how the data were extracted. I do hope the authors can look at these questions in the future. I have no further comments.
Author Response
The authors have addressed all my previous comments satisfactorily. I realize that the request to examine correlations between novelty preferences and attention toward AOIs as a function of familiarization pose is not possible due to the way the study was developed and/or how the data were extracted. I do hope the authors can look at these questions in the future. I have no further comments.
Thank you for all your feedback! We hope that we will be able to reconstruct the data regarding the poses and publish the results at some point in the future.